# The Emerging Role of Immunoglobulins and Complement in the Stimulation of Neuronal Activity and Repair: Not as Simple as We Thought

**DOI:** 10.3390/biom14101323

**Published:** 2024-10-18

**Authors:** Tatyana Veremeyko, Natasha S. Barteneva, Ivan Vorobyev, Eugene D. Ponomarev

**Affiliations:** 1Department of Biology, School of Sciences and Humanities, Nazarbayev University, Astana 010000, Kazakhstan; veremeyko.tat@gmail.com (T.V.); natalie.barteneva@nu.edu.kz (N.S.B.); ivan.vorobyev@nu.edu.kz (I.V.); 2Laboratory of Cell Motility, National Laboratory Astana, Nazarbayev University, Astana 010000, Kazakhstan; 3Department of Biomedical Sciences, Jockey Club College of Veterinary Medicine and Life Sciences, City University of Hong Kong, Hong Kong

**Keywords:** central nervous system (CNS), neurodegeneration, CNS repair, complement, immediate early genes (IEGs), immunoglobulins, natural autoantibodies (NAAs), B1 B cells

## Abstract

Neurologic disorders such as traumatic brain injury, multiple sclerosis, Alzheimer’s disease, and drug-resistant epilepsy have a high socioeconomic impact around the world. Current therapies for these disorders are often not effective. This creates a demand for the development of new therapeutic approaches to treat these disorders. Recent data suggest that autoreactive naturally occurring immunoglobulins produced by subsets of B cells, called B1 B cells, combined with complement, are actively involved in the processes of restoration of neuronal functions during pathological conditions and remyelination. The focus of this review is to discuss the possibility of creating specific therapeutic antibodies that can activate and fix complement to enhance neuronal survival and promote central nervous system repair after injuries associated with many types of neurodegenerative diseases.

## 1. Introduction

An important role of natural autoantibodies (NAAs) as potential therapeutic agents historically arises from applications of intravenous immunoglobulin (IVIG) fractions isolated from the serum of healthy human adult subjects. IVIG is a sterile product used for non-specific therapy in neurology, hematology, immunology, nephrology, and dermatology [1]. In dermatology, IVIG is used to treat Kawasaki disease (vasculitis), a disorder recently found to be associated with COVID-19 in children [2]. The most notable results for the usage of IVIG therapy were achieved for the treatment of several neurologic disorders, including traumatic brain injury [3] and multiple sclerosis [4]. A substantial decrease in the plasma levels of pathogenic Aβ_42_ peptides was also observed in IVIG-treated patients with Alzheimer’s disease [5]. Currently, the mechanism of action of IVIG is not well understood, and, mechanistically, it is associated with the deactivation of antigen-presenting cells, such as macrophages and dendritic cells, via Fc receptors (FcR) in an antigen-independent manner [1]. It is also suggested that the therapeutic action is mostly mediated by IgG antibodies representing ~90–95% of all immunoglobulins in the serum of healthy donors. However, concentrations of other isotypes of immunoglobulins such as IgA and IgM significantly vary from one donor to another [1]. In healthy human subjects, IgM and IgA represent, on average, less than 5–10% of all immunoglobulins, reaching the peak by the age of 30–40 years, followed by a sharp decline during aging [6]. Most IgM and IgA isotypes in the serum of normal subjects represent NAAs, which have broad spectrum specificity and are often self-reactive, binding to carbohydrates and lipids on the surface of damaged or apoptotic cells. It is suggested that these IgM or IgA immunoglobulins recognize known or unknown self-antigens and mediate a therapeutic effect via the Fc portion of antibodies that bind recently acknowledged FcRs for IgM and low-affinity FcRs for IgA (known as FcµR) and both IgM and IgA (known as FcαµR) [7], and by binding the Fab portion of antibodies to a self-antigen on the surface of damaged or normal cells [2]. In particular, it was shown that self-reactive IgM could bind to the surface of oligodendrocytes and their progenitors (cells that comprise myelin sheath in the central nervous system (CNS)) and promote re-myelination after CNS injury acting via FcμR and FcαμR [8]. The antigen specificity of these antibodies is still poorly characterized, and they are usually present in the serum of healthy donors in low concentrations, which explains the limited therapeutic efficacy of IVIG. To understand the specificity and properties of therapeutic antibodies in IVIG, it is important to understand the basic biology of the specific B cell clones that produce these therapeutic self-reactive IgM or IgA within the CNS. During development, most NAAs are produced by a specific type of B cells called B1 B cells [9]. After birth, the percentage of B1 B cells in the peripheral blood gradually declines over time, reaching a minimum level during the aging process similar to the level of NAAs [6,10]. During most of adulthood, B1 B cells remain the main source of NAAs that have therapeutic potential in humans [11]. In this review, we will discuss possible antigen specificity and the mechanism of action of NAAs that can activate and fix complement on the surface of neuronal cells to influence the function of neurons, including their synaptic activity, response to neurologic insult, survival, and repair.

## 2. B1 B Cells

### 2.1. Mouse B1 B Cells

B1 B cells are a subset of innate-like B lymphocytes that are older than conventional B cells (referred to as B2 B cells). B1 cells are phenotypically distinguished from B2 cells by the expression of a panel of specific markers, early appearance during development, and homing in on specific locations in body cavities such as the peritoneal, pleural, and CNS ventricles [12]. For example, in the peritoneum, ~70% of all B cells are B1 B cells. Mouse peritoneal B1 B cells express the marker CD11b, which makes these cells similar to innate immune cells, such as macrophages and myeloid dendritic cells, which also express this marker. In contrast to body cavities, B1 B cells are found in the secondary lymphoid organs such as the spleen and lymph nodes at very low frequencies (1–2% and 0.2% of all B cells, respectively) [13]. B1 B cells mostly produce IgM and rarely switch to other classes of antibodies (e.g., IgA or IgG3) compared to the conventional B2 B cells that often switch to IgG1. Finally, B1a B cells are the only type of B lymphocytes that could co-express the CD11b macrophage and the pan-T-cell marker CD5. In mice, B1a cells are conventionally referred to as CD45R^low^CD23^low^CD19^+^IgM^+^CD5^+^ cells. In the peritoneal cavity, B1 B cells express the earlier mentioned marker CD11b, which is absent in B1 B cells in the spleen, lymph nodes, or peripheral blood. Thus, B1 B cells can be CD11b-positive or CD11b-negative. At the same time, B2 B cells are CD45R^hi^CD23^+^CD19^+^IgD^+^IgM^variable^CD5^−^. Although B1 and B2 cells are not always easy to distinguish in tissues at the site of inflammation, B1a B cells can be distinguished from B2 cells such as IgM^hi^CD5^+^ [13,14].

### 2.2. Human B1 B Cells

B cell subsets isolated from immunodeficient patients have the CD19^hi^CD21^low^CD23^−^CD86^hi^IgM^hi^ phenotype and have many phenotypic and functional similarities with mice B1 B cells [15]. More recent studies suggest the identification of human B1 B cells as CD19^+^CD20^+^CD27^+^CD38^low/int^CD43^+^ [10]. Human B1 B cells are the most attractive targets for treating neurodegenerative diseases as they can produce regulatory NAAs that promote tissue repair in humans [11].

### 2.3. Function of B1 B Cells and Their Antibodies

Despite intensive investigations for many decades, the main function of B1 B cells remains unclear. It is known that these cells are responsible for the production of naturally occurring germline-encoded antibodies (referred to as NAAs in this review), most of which are IgM, on rare occasions are IgA, and even more rarely are IgG3 [9]; however, as innate-like cells, B1 B cells can also play an important role in debris or apoptotic cell clearance, antigen presentation, and the production of immunoregulatory cytokines such as IL-10 [16,17]. Yet, so far, the main function of B1 B cells is mostly connected with the function of B1-derived NAAs. The exact specificity of these antibodies is not well known for most B1 B cell clones, and their autoimmune nature is related to pathological conditions. In addition, it was shown that these antibodies often recognize carbohydrate and lipid self-antigens [14,18]. It was initially assumed that B1 cell-derived autoimmune IgMs that bind to gangliosides contribute to autoimmune diseases affecting peripheral nerves known as Guillain-Barre syndrome (GBS) [19]. However, more recent studies suggested that GBS is mostly connected with IgG1 but not IgM, IgA, or IgG3 autoantibodies, indicating that this disease is most likely mediated by conventional B2 rather than B1 B cells [20,21]. Recent data also indicate that IgM promotes CNS repair by binding to oligodendrocytes, promoting their expansion and differentiation, and guiding neurite outgrowth [8,22], thus confirming the therapeutic potential of B1 B cells and their NAAs. The specificity of most common NAAs that recognize brain-specific antigens is summarized in Table 1.

It is quite interesting to notice that many antigens for NAAs in the CNS are located in neuronal lipid rafts (NLRs), which are located in post-synaptic membranes that are enriched in neurotransmitter receptors and brain-specific gangliosides at the area of the synaptic cleft [23]. These antigens are associated with many types of pathologies, including traumatic brain injury, Alzheimer’s disease, multiple sclerosis, and epilepsy [24,25,26,27]. How NLRs could induce the production of NAAs is currently not clear, but we have previously shown that during blood–brain barrier (BBB) disruption, platelets can recognize brain-specific gangliosides and initiate neuroinflammation, which can possibly lead to activation of B1 B cells among other immune cells such as microglia, macrophages, and T cells [23,24,28]. Currently, the role of B1 B cells and their NAAs in the regulation of neuroinflammation is not well-known. B1-derived IgMs could constitute an active component of IVIG, which requires further investigation. Even though IgM is present at low concentrations in IVIG [1], the identification of therapeutic B1 B cell clones that mediate the downregulation of neuroinflammation and promote CNS repair would allow the synthesis of monoclonal antibodies with a high level of efficiency to treat acute neurologic disorders. The role of these antibodies in particular disorders is discussed below.

**Table 1 biomolecules-14-01323-t001:** Antigen specificity of common NAAs that bind to CNS-specific antigens.

Antigen	Location in the CNS	Isotype (Clone)	Species (Source of Ab ^1^)	Relation to Pathology and/or Repair	References
AQP4 ^2^	Astrocytes (limitans)	IgG	Human	Neuromyelitis Optica	[29]
NMDAR ^3^	Neurons (NLR ^4^)	IgM, IgA	Human	Encephalitis, psychosis, seizures	[30]
AMPAR ^5^	Neurons (NLR)	IgG, IgM	Human	Encephalitis, seizures	[30]
GABA_A_R ^6^	Neurons (NLR)	IgG1, IgG3, IgM	Human	Encephalitis, psychosis, seizures	[31]
D2R ^7^	Neurons (NLR)	IgG, IgM	Human	Parkinson’s disease, psychosis	[32]
MOG ^8^	Oligodendrocytes	IgM	Mice	Multiple sclerosis	[33]
Sulfatide	Oligodendrocytes	IgM (hIgM22)	Human	Remyelination	[34]
Gangliosides	Neurons (NLR)	IgM (sHIgM12)	Human	Axonal outgrowth	[35]
Amyloid	Neurons (NLR)	IgM	Human	Alzheimer’s disease, neuroprotection	[36]
MLD ^9^	Neurons, astrocytes	IgM	Human	Schizophrenia	[37]
PC ^10^	CNS (including NLR)	IgM	Human	Stroke, Alzheimer’s disease	[38,39]

^1^ Ab, antibodies; ^2^ AQP4, aquaporin-4; ^3^ NMDR, N-methyl-D-aspartate receptor; ^4^ NLR, neuronal lipid rafts; ^5^ AMPAR, α-amino-3-hydroxy-5-methyl-4-isoxazolepropionic acid receptor; ^6^ GABA_A_R, γ-aminobutyric acid type A receptor; ^7^ D2R, dopamine type 2 receptor; ^8^ MOG, myelin oligodendrocyte glycoprotein; ^9^ MLD, malondialdehyde; ^10^ PC, phosphorylcholine.

## 3. Traumatic Brain Injury

Traumatic brain injury (TBI) is one of the leading causes of death and disability among young adults worldwide, with a high socioeconomic impact [40]. TBI could also predispose patients to the development of other neuropathological conditions such as Alzheimer’s disease, amnesia, and epilepsy [41]. The role of B1 B cells in TBI is unclear and has not been specifically investigated so far. It was reported that certain TBI patients could develop antibodies against gangliosides, but the role of these antibodies in TBI pathology is unclear [42]. We found that *st3gal5*-deficient mice that lack the major brain gangliosides GM1, GM3, GT1b, and GQ1b demonstrate less inflammation in the CNS after TBI, but more extensive neuronal damage and cognitive decline [24]. This indicated an important role of brain gangliosides in the development of CNS inflammation and neuronal repair as a potential target for B1 cells. However, the functions of B1 B cells as potential sources of anti-ganglioside antibodies have not been extensively investigated in the TBI models. Thus, the role of B1 B cells and B1-derived IgM or IgA in TBI remains unclear; however, the activation of B cells and the production of self-reactive antibodies were reported to be a hallmark of TBI or spinal cord injury [43,44,45], indicating the possibility of stimulating CNS repair after TBI using self-reactive antibodies such as sHIgM12 that bind to brain-specific gangliosides or other CNS autoantigens and promote CNS repair (Table 1). Since it is known that autoantibodies that bind to neuronal cells could activate complement [46], it is important to detect and investigate the role of complement in neurologic disorders. In humans with TBI or animal models, the levels of complement subunits C1q, C3b, and C3d were elevated in the area of injury [47,48]. C3- and C4- (but not C1q-) deficient mice demonstrated decreased areas of injury and milder motor function deficits, indicating the pathological roles of both C3 and C4 [49,50].

## 4. Multiple Sclerosis

Multiple sclerosis (MS) is a chronic autoimmune disease of the CNS that predominantly affects young adults, leading to substantial neurological disability that includes upper and lower motor syndrome. MS and experimental autoimmune encephalitis (EAE, an animal model for MS) involve autoimmune T cells that recognize myelin self-antigens, such as myelin oligodendrocytes glycoprotein (MOG) [51]. MS onset usually starts as relapsing-remitting MS (RRMS), which is characterized by multiple relapses followed by spontaneous remission. RRMS eventually transforms into secondary-progressive MS. Recently, the role of B cells has been recognized in MS pathology. B cells could play a pathological role in MS by presenting self-antigen (e.g., MOG, see Table 1), stimulating pathogenic T cells, and producing autoantibodies that bind to myelin self-antigens on the surface of oligodendrocytes and activate complement, contributing to the destruction of oligodendrocytes. However, it was also proposed that B cells could be beneficial for MS, serving as regulatory B cells (Bregs) via the production of IL-10 (and possibly other regulatory cytokines IL-35 and TGFβ) that suppress CNS inflammation [52,53]. At the same time, the role of B1 B cells and B1-derived self-reactive NAA in MS is not well understood; however, it was demonstrated that the level of B1 B cells decreases during MS progression. We have previously found that antibodies to ganglioside GQ1b (abundant in NLR) dramatically decreased EAE severity, indicating the important role of anti-ganglioside autoantibodies in controlling neuroinflammation and/or preserving neuronal functions [28]. Moreover, certain human autoantibodies, such as the hIgM22 clone, promote remyelination and could be used for future MS therapy (Table 1). In cortical inflammatory lesions of patients with MS, C1q, C3b, and C3d complement subunits were detected [54,55,56]. However, in EAE models, C1q- and C3-deficient animals did not demonstrate significant changes in disease scores [57]. Thus, the role of complement in MS remains unclear.

## 5. Alzheimer’s Disease

Alzheimer’s disease (AD) is a type of dementia associated with cognitive decline and is commonly associated with aging. Pathologically, AD is characterized by the accumulation of aggregated amyloid and tau proteins, synaptic dysfunction, and brain atrophy [58]. The role of B cells in the pathology of AD remains controversial. A recent study suggests that regulatory B cells ameliorate AD-associated neuroinflammation by producing regulatory cytokine IL-35 [59]. Another study suggests that the depletion of B cells using anti-CD20 antibodies or B cell-deficient mice resulted in the amelioration of AD in three different mouse models of AD that included 3×TgAD, APP/PS1, and 5×FAD mice [60]. It is interesting that in the study mentioned above, the numbers of B1a and B1b cells were increased in the CNS of the 3×TgAD transgenic mouse model of AD. However, in the 5×FAD model, the numbers of B1a and B1b cells were not significantly different from the non-transgenic (control) group of mice, indicating model-specific differences in the different mouse models of AD [60]. Thus, the therapeutic effect described in this study was likely due to the depletion or knockout of conventional B2 rather than B1 cells. Indeed, studies in mice and humans indicate a decrease in the number of B1 B cells during aging [61] when AD becomes the prevailing disease. In contrast to the pathological role of B cells, there is more evidence that B1-derived NAAs play a protective role in AD [5,36,39]. Our study indicated that a deficiency in brain-specific gangliosides that are predominantly located within NLRs significantly ameliorated AD pathology in the 5×FAD mouse model of AD. Moreover, the treatment of mice with sialic acid-binding lectin resulted in a substantial improvement in AD pathology by lowering amyloid burden, brain atrophy, and cognitive decline [25]. Since sialic acid is an external part of many types of brain-specific gangliosides and often serves as an external epitope for anti-ganglioside antibodies, this could indicate that NAAs with specificity to gangliosides can be used as future therapeutic agents for AD. In addition, it was also found that NAAs with specificity to phosphorylcholine (also located in NLR) and amyloid significantly ameliorated AD pathology (Table 1). In vitro, C1q protected neurons from Aβ toxicity, while APP-transgenic mice exhibited decreased glial activation markers with no decrease in plaque load [62]. On the other hand, C3-deficient APP transgenic mice had elevated amyloid burden but improved cognitive functions compared to wild-type transgenic animals [63].

## 6. Epilepsy

Epilepsy is a disease characterized by uncontrolled seizures and a disbalance of activating vs. inhibitory neurotransmitters and/or their receptors. Although treatments for epilepsy are available, about 30% of patients do not respond to available drugs that modulate the function of activating or inhibitory neurotransmitter receptors [64]. Epilepsy is associated with an inflammatory response in the CNS and possible involvement of B cells, their antibodies, and complement [65]. The detection of NAAs that recognize activating and inhibitory neurotransmitter receptors (NAMDR and GABA_A_R, respectively) was associated with the development of epileptic seizures, as shown in Table 1. Although the level of Bregs was decreased in patients with epilepsy [66], this could indicate systemic changes in balance toward the development of neuroinflammation [67]. The binding of NAAs to NLR could change signaling downstream of neurotransmitter receptors and the levels of neuronal electric activity, which may have beneficial or detrimental effects on the development of seizures. The serum levels of C3 and C4 were elevated in animal models of epilepsy [68]. C1q-and C4-deficient mice (but not C3-deficient mice) exhibit higher levels of epileptic seizures when compared to wild-type control animals, suggesting that decreased synaptic pruning at certain developmental periods could lead to excessive excitotoxicity [69,70].

## 7. Emerging Role of B1 B Cells and NAAs in Neurological Diseases

For a very long time, the role of B1 B cells in the development of neurological diseases was unknown. However, recent research has shown that B1a cells accumulate in normal CNSs during development. These cells were CD45^+^CD19^+^CD45R^+^IgM^+^CD5^+^ and they were found in the CNS at high frequencies in the brains of developing P1-P21 (post-natal day 1 to day 21) mice, but not in adult 8-week-old animals [71]. The authors of this study demonstrated that these cells were located in brain ventricles, and their IgM played an important role in binding to oligodendrocyte progenitors via the specific receptor for IgM (FcμR) and stimulating the proliferation of oligodendrocyte progenitors and their differentiation into mature oligodendrocytes [71]. FcμR, also known as CD351, is an Fc receptor that binds IgM with a very high affinity and intermediate affinity for IgA, both of which are produced by B1 B cells. Thus, B1a cells play an important role in myelination via producing self-reactive IgM, which may also be important for CNS repair after injury and blood–brain barrier (BBB) disruption, which is associated with MS or TBI. However, the specificity of most self-reactive IgM and IgG antibodies is not known. Understanding the specificity of B1-derived self-reactive IgM is very important for creating highly effective and specific therapeutic antibodies, which would replace current IVIG therapy that uses whole immunoglobulin fractions from healthy donors.

## 8. New Role of Complement in Neurologic Diseases

The complement system consists of ~30 proteins that comprise an enzymatic self-amplification cascade initiated by carbohydrates or lipids of pathogens, cellular debris, and blood coagulation cascade. There are three paths of complement activation: the classical, lectin, or alternative pathways [46,72,73]. The classical pathway is triggered by the formation of antigen-antibody complexes that bind C1q in the C1 complex, leading to the proteolytic cleavage of C2 into C2a, and C2b and C4 into C4a and C4b. Subsequentially, C4b and C2b form the C4bC2b complex (C3-convertase), which cleaves C3 into C3a and C3b. The lectin pathway is activated by carbohydrate antigens that serve as ligands for mannose-binding lectin. This pathway initiates the same cascade as the classical pathway, leading to the formation of C3a and C3b. The alternative pathway is not activated by antigen-antibody complexes. This pathway is constitutively active at a low level and is initiated by the spontaneous hydrolysis of C3, which then binds to proenzyme Factor B (fB), enabling the cleavage of fB to fBb by Factor D (fD) and subsequent cleavage of C3 into C3a and C3b. C3b then binds to Factor B, which is cleaved by fD, forming the C3 convertase C3bBb that cleaves more C3 into C3a and C3b [46,72]. During blood coagulation, activated factor XII (FXIIa) activates the classical complement pathway via interaction with C1q [74,75]. Moreover, thrombin, kallikrein, and plasmin directly cleave C3, leading to the formation of C3a and C3b [76]. Active C3b, downstream of all pathways, leads to the cleavage of C5 (into C5a and C5b), the formation of the membrane attack complex (MAC), and lysis of pathogens or target cells. The fragments C3a and C5a activate mast cells and macrophages and thus play a pro-inflammatory role. The active enzymes C3b and C4b are later degraded by factor I (FI) in the presence of other co-factors, such as complement receptor CR1, via a two-step process wherein FI and co-factors cleave C3b to make an iC3b fragment, which is then cleaved into C3c and C3d. Similarly, C4b is cleaved by FI into C4c and C4d [72,77]. C3c and C4c have similar sizes and structures, and their functions are not well understood. The liver is the dominant source of most complement proteins in the plasma, except C1q, which is synthesized in leukocytes [78].

In the CNS, most complement components, including C1q, C4, and C3, are synthesized by astrocytes, oligodendrocytes, neurons, and microglia [79]. Platelets can also secrete C3 [28], and these cells play a key role in the initiation of neuroinflammation after TBI [16]. Microglia express activating complement receptors CR1, CR3, and CR4 [80], while neurons express inhibitory receptors and recently discovered receptors activating CSMD [81]. Recent studies demonstrated the involvement of the complement system in the pathogenesis of TBI, AD, amyotrophic lateral sclerosis (ALS), and MS [77,82,83,84]. An association was also observed between C4 gene polymorphism and schizophrenia [85,86], similar to anti-malondialdehyde IgM (Table 1). In the normal CNS, microglia produce C1q, which can bind to “weak” synaptic membranes (via exposed components such as phosphatidyl serine). When C1r and C1s are complexed with C1q to form C1, activation will cleave C4 and C3, resulting in covalent binding to axons that are marking synaptic areas for elimination by microglia via CR3. [87,88]. However, it is currently not clear whether antibodies bind to damaged neurons and/or dysfunctional synapses to fix C1 and activate the C1-C2-C4-C3 pathway to eliminate damaged or dysfunctional synapses. In our study, we found that C4-deficient mice were susceptible to pentylenetetrazol (GABA_A_R antagonist)-induced seizures, leading to a very high mortality rate [70]. Interestingly, in our study, C3-deficient mice had slightly milder seizure scores [70], which was consistent with other studies indicating the pathogenic role of C3 in epilepsy [89,90]. We found that during seizures, C4-, but not C3-, deficient mice were unable to upregulate immediate early response genes (IEGs) in the CNS, including neuronal Egr1 and Npas4 and their downstream neuronal survival/repair factors such as BDNF, which lead to a high mortality rate and cognitive problems [70]. Thus, our data indicate that C4 plays a role in other functions besides activating C3 and promoting synapse pruning. We found that C4 contributed to neuronal activation via the induction of IEGs.

As the new mechanism of action of C4 in the CNS, we propose that NAAs produced by B1 B cells bind to self-antigens in the area of damaged synapses that are enriched with NLR. B1-derived IgM or IgA autoantibodies could enter normal CNSs via transcytosis from CNS ventricles where B1 B cells reside or could enter CNSs from blood vessels during pathological conditions such as TBI, MS, AD, or epilepsy when the BBB becomes compromised. Moreover, B1 B cells accumulate in the CNS during neuroinflammation associated with neurological disorders [91]. Intrathecal synthesis of immunoglobulins is now recognized as the hallmark of many neurological disorders [92]. After the binding of IgM to neuronal lipid rafts on the post-synaptic membrane, IgM recruits C1 and C4 to form unified IgM-C1-C4b complexes, as was demonstrated by [93]. The formation of these complexes has important functions for neurons and glia. This includes (1) the modulation of signaling downstream of neurotransmitter receptors; (2) engaging signaling downstream of gangliosides or other carbohydrate/lipid antigens; (3) engaging complement receptor inhibitors on neuronal cells (e.g., CSMD family of activating complement receptors or CD59 [94,95]) or microglia (e.g., CR1 complement receptors); (4) engaging FcμR receptors on oligodendrocyte progenitors and microglia to stimulate remyelination and tissue repair and remodeling; and (5) induction of IEGs (e.g., Egr1 and Npas4) and its downstream neurotrophic factors (e.g., BDNF) leading to enhanced neuronal survival during CNS insult and its subsequent repair (Figure 1). Limited information is available about the formation of immunoglobulin and complement complexes in the CNS due to methodological problems in detecting them in real-time in the brain. However, some studies detect these complexes in brain blood vessels, indicating an important role of these complexes in the therapy of neurological disorders such as AD [96].

## 9. Conclusions and Future Directions

Traditionally, autoantibodies with complement were considered contributors to neurologic disorders by enhancing inflammation and damaging the membranes of target cells by MAC. However, recent data indicate that specific clones of B cells (e.g., IgM-producing B1 B cells) and certain complement subunits (e.g., C4) also play neuroprotective roles and could be used in the future therapy of many types of neurologic disorders, including TBI, MS, AD, and epilepsy. We hypothesize that the beneficial rather than detrimental action of NAAs is connected with the antigen specificity of these antibodies directed against neuronal cells in general and NLR in particular (see Table 1). At the synapse area, neurons express potent complement inhibitors that prevent the formation of MAC and subsequent membrane damage. Future research will focus on the investigation of antigen specificity of particular B1 B cell clones and the role of IgM-C1-C4b complexes in the modulation of neuronal functions.

## Figures and Tables

**Figure 1 biomolecules-14-01323-f001:**
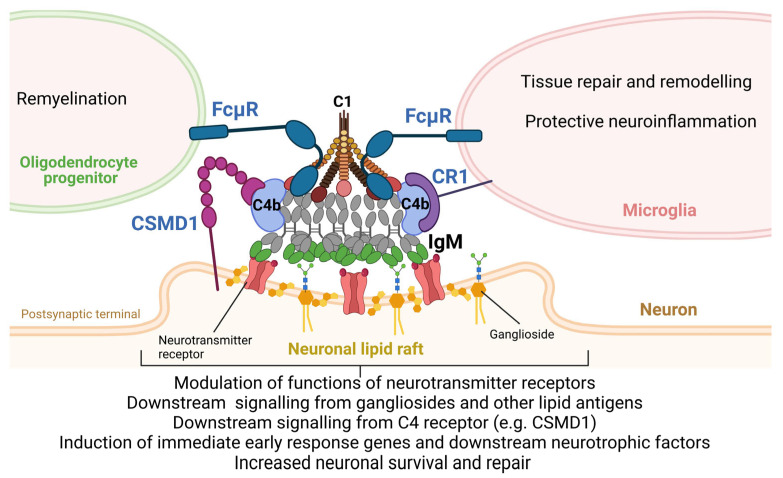
The proposed model of regulation of the CNS repair process by B1-derived NAAs (IgM) and complement subunits C1 and C4 by forming IgM-C1-C4b complexes that affect neuronal cells, microglia, and oligodendrocyte progenitors. The image was created with BioRender.com.

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
