# Peer review of "The Emerging Role of Immunoglobulins and Complement in the Stimulation of Neuronal Activity and Repair: Not as Simple as We Thought"

_biomolecules, 2024, doi:10.3390/biom14101323_

Round 1
Reviewer 1 Report
Comments and Suggestions for Authors
This review highlights the work of Ponomarev providing an Interesting perspective of the role for B1 B cells in CNS disorders. This is a currently evolving area for research that includes the characterization of antibodies (NAA) present under nonpathologic conditions that may have neuroprotective roles. In addition, given the current therapeutic strategies targeting the early components of the classical complement pathway now in clinical trials, the evidence for potential neuroprotective roles of these components highlighted in this manuscript are important to be considered. Thus, the review is timely. However, in addition to the grammatical errors there are serious inaccuracies and/or imprecise statements in the discussion of the complement cascade that must be revised.
Line 252: add in “IgM and IgG antibody”
Line 254: “…antigen-antibody complexes “bind C1q in the C1 complex” leading to the …”
Line 261 “hydrolysis of C3 (delete “into C3a and C3b”) which then binds to proenzyme Factor B (fB) enabling the cleavage of fB to fBb by Factor D (fD), and subsequent cleavage of C3 into C3a and C3b. C3b then binds Factor B which is cleaved by fD forming the C3 convertase C3bBb that cleaves more C3 into C3a and C3b.”
Line 264: This reviewer cannot find a reference supporting the cleavage of “C1q” by FXIIa, and even if it did would not lead to the cleavage of C4 and C2. This whole section must be rewritten and properly referenced.
A current review of the complement system that should be helped is by Mastellos et al., 2024 (Mastellos et al., 2024).
Line 271: (CR)-1 should be “CR1”.
Lines 284-287: Change C1q to C1 as C1q must be in the C1 complex for “activation” of the classical pathway to tag synapses. Also add C2 to line 287.
Comments on the Quality of English LanguageLine 85 and should be “or”
Line 168 and should be “cells”
Line 184 “and very perspective” revise for sense
Line 190 disfunction should be “dysfunction”
Line 193 Insert “.” As needed
Line 225 Binging should be “Binding”
Line 236 verticals should be ???
Line 305 when should be “where”
Author Response
This review highlights the work of Ponomarev providing an interesting perspective of the role for B1 B cells in CNS disorders. This is a currently evolving area for research that includes the characterization of antibodies (NAA) present under nonpathologic conditions that may have neuroprotective roles. In addition, given the current therapeutic strategies targeting the early components of the classical complement pathway now in clinical trials, the evidence for potential neuroprotective roles of these components highlighted in this manuscript are important to be considered. Thus, the review is timely. However, in addition to the grammatical errors there are serious inaccuracies and/or imprecise statements in the discussion of the complement cascade that must be revised.
Response: We revised statements related to complement cascade and performed additional proofreading to fix grammatical errors. The main changes are marked in bold.
Line 252: add in “IgM and IgG antibody”
Response: We have made this correction.
Line 254: “…antigen-antibody complexes “bind C1q in the C1 complex” leading to the …”
Response: We have made this correction.
Line 261 “hydrolysis of C3 (delete “into C3a and C3b”) which then binds to proenzyme Factor B (fB) enabling the cleavage of fB to fBb by Factor D (fD), and subsequent cleavage of C3 into C3a and C3b. C3b then binds Factor B which is cleaved by fD forming the C3 convertase C3bBb that cleaves more C3 into C3a and C3b.”
Response: We have made the proposed changes.
Line 264: This reviewer cannot find a reference supporting the cleavage of “C1q” by FXIIa, and even if it did would not lead to the cleavage of C4 and C2. This whole section must be rewritten and properly referenced.
Response: We changed this section and included references.
A current review of the complement system that should be helped is by Mastellos et al., 2024 (Mastellos et al., 2024).
Response: We added this review as a reference.
Line 271: (CR)-1 should be “CR1”.
Response: We have made this correction.
Lines 284-287: Change C1q to C1 as C1q must be in the C1 complex for “activation” of the classical pathway to tag synapses. Also add C2 to line 287.
Response: We have made the proposed changes.
Comments on the Quality of English Language
Line 85 and should be “or”
Response: We have made this correction.
Line 168 and should be “cells”
Response: We have made this correction.
Line 184 “and very perspective” revise for sense
Response: We have made this revision.
Line 190 disfunction should be “dysfunction”
Response: We have made this correction.
Line 193 Insert “.” As needed
Response: We have made this correction.
Line 225 Binging should be “Binding”
Response: We have made this correction.
Line 236 verticals should be ???
Response: We corrected it to “ventricles”
Line 305 when should be “where”
Response: We have made the proposed change.

Reviewer 2 Report
Comments and Suggestions for Authors
The article entitled “The emerging role of immunoglobulins and complement in the stimulation of neuronal activity and repair: not as simple as we thought,” want to show the possibility of creating specific therapeutic antibodies that can activate and fix complement to enhance neuronal survival and promote central nervous system repair after neurodegenerative diseases. However, there still some main points for improvement or further consideration.
1. The review want to discuss the emerging role of immunoglobulins and complement. However, the manuscript mainly talk about the function of B1 B cells and their antibodies and the effect on the neurodegenerative diseases. Why not discuss detail effect of complement or complexes (immunoglobulins and complement) on the specific neurodegenerative diseases?
2. autoantibodies with complement were considered contributors to neurologic disorders by enhancing inflammation and damaging membranes of target cells. However, recent data indicates that specific clones of B cells and certain complement subunits also play neuroprotective roles. Why is there so big difference? I think the authors should discussion.
3. The author talk about emerging role of immunoglobulins and complement together. How many kinds immunoglobulins and complement complexes reported? Detail function of complexes on the CNS?
4. Many punctuation marks are missing in the text.
Author Response
The article entitled “The emerging role of immunoglobulins and complement in the stimulation of neuronal activity and repair: not as simple as we thought,” want to show the possibility of creating specific therapeutic antibodies that can activate and fix complement to enhance neuronal survival and promote central nervous system repair after neurodegenerative diseases. However, there still some main points for improvement or further consideration.
Response: In the revised version of the manuscript, we addressed specific points as indicated below. The main changes are marked in bold.
- The review want to discuss the emerging role of immunoglobulins and complement. However, the manuscript mainly talk about the function of B1 B cells and their antibodies and the effect on the neurodegenerative diseases. Why not discuss detail effect of complement or complexes (immunoglobulins and complement) on the specific neurodegenerative diseases?
Response: In the revised version we added paragraphs that discuss the role of complement in particular neurologic disorders (TBI (line 169), MS (line 197), AD (line 230), and epilepsy (line 249)).
- Autoantibodies with complement were considered contributors to neurologic disorders by enhancing inflammation and damaging membranes of target cells. However, recent data indicates that specific clones of B cells and certain complement subunits also play neuroprotective roles. Why is there so big difference? I think the authors should discuss.
Response: It is a good and important point. Very little is known about the exact mechanisms. We hypothesize that the beneficial rather than detrimental action of NAAs is connected with the antigen specificity of these antibodies directed against neuronal cells. CNS-resident cells express potent complement inhibitors (e.g. CD59, CSDM1) that prevent the formation of MAC and membrane damage. We added these points to the Conclusion section, line 360.
- The author talk about emerging role of immunoglobulins and complement together. How many kinds immunoglobulins and complement complexes reported? Detail function of complexes on the CNS?
Response: Limited information is available about the formation of immunoglobulin and complement complexes in the CNS due to methodological problems in detecting them in real-time in the brain. Some studies detect these complexes in brain blood vessels. We added this information in the revised version of the manuscript, line 347.
- Many punctuation marks are missing in the text.
Response: We performed additional proofreading to correct punctuation and other grammatical errors.

Round 2
Reviewer 1 Report
Comments and Suggestions for Authors
The authors, while inserting noted corrections, still got some of the complement sections wrong. I corrected them on the authors draft: largely sections 5, 6, and 8. Attached here.

Should be reviewed once more by an English editor.
Author Response
The authors, while inserting noted corrections, still got some of the complement sections wrong. I corrected them on the authors draft: largely sections 5, 6, and 8. Attached here.
Response: we included proposed corrections in sections 5, 6 and 8. Marked in bold.
Reviewer 2 Report
Comments and Suggestions for Authors
I have no more question.
Author Response
We than Reviewer for valuable suggestions.